# Establishing the Competency Development and Talent Cultivation Strategies for Physician-Patient Shared Decision-Making Competency Based on the IAA-NRM Approach

**DOI:** 10.3390/healthcare10101844

**Published:** 2022-09-23

**Authors:** Shan-Fu Yu, Chih-Ming Hsu, Hui-Ting Wang, Tien-Tsai Cheng, Jia-Feng Chen, Chia-Li Lin, Hsing-Tse Yu

**Affiliations:** 1Division of Rheumatology, Allergy, and Immunology, Department of Internal Medicine, Kaohsiung Chang Gung Memorial Hospital, Kaohsiung City 833, Taiwan; 2Division of Rheumatology, Allergy, and Immunology, Department of Internal Medicine, Chiayi Chang Gung Memorial Hospital, Puzi City 613, Taiwan; 3School of Medicine, College of Medicine, Chang Gung University, Tayouan City 333, Taiwan; 4Graduate Institute of Adult Education, National Kaohsiung Normal University, Kaohsiung City 802, Taiwan; 5Medical Education Department, Chiayi Chang Gung Memorial Hospital, Puzi City 613, Taiwan; 6Department of Business Administration, National Chung Cheng University, Minxiong Township, Chiayi 621, Taiwan; 7Department of Emergency Medicine, Kaohsiung Chang Gung Memorial Hospital, Kaohsiung City 833, Taiwan; 8Department of International Business, Ming Chuan University, Taipei City 111, Taiwan; 9Department of Obstetrics and Gynecology, Taipei Chang Gung Memorial Hospital, Taipei City 105, Taiwan

**Keywords:** shared decision making, physician, competency, IAA-NRM, DEMATEL

## Abstract

Shared decision making (SDM) is a collaborative process involving patients and their healthcare workers negotiating to reach a shared decision about medical care. However, various physician stakeholders (attending physicians, medical residents, and doctors in post-graduate years) may have different viewpoints on SDM processes. The purpose of this study is to explore the core competence of physicians in performing SDM tasks and to investigate the significant competency development aspects/criteria by applying the literature research and expert interviews. We adopt the IAA (importance awareness analysis) technique for different stakeholders to evaluate the status of competency development aspects/criteria and to determine the NRM (network relation map) based on the DEMATEL (decision-making trial and evaluation laboratory) technique. The study combines the IAA and NRM methods and suggests using the IAA-NRM approach to evaluate the adoption strategies and common suitable paths for different levels of physicians. Our findings reveal that SDM perception and practice is the primary influencer of SDM competence development for all stakeholders. The current model can help hospital administrators and directors of medical education understand the diverse stakeholders’ perspectives on the core competence of SDM tasks and determine common development plans. It provides strategic directions for SDM competency development and talent cultivation programs.

## 1. Introduction

Shared decision making (SDM) is an interactive communication process that involves patients and their healthcare professionals working jointly to make decisions based on the most relevant evidence-based information, weighted by the patients’ preferences, beliefs, and values [1]. Compared with the traditional paternalistic doctor–patient relationship, SDM is a new model of decision making based on the concept of patient-centered care, which has been vigorously promoted since the late 1980s. Charles et al. [2] first proposed the theoretical framework for SDM in 1997. It is the most widely accepted model suggesting that both the doctor and patient are involved, sharing information, reaching a consensus, and agreeing with opinions for implantation. Through mutual discussion, exchange of information, and respect for patient autonomy, this collaborative process enables patients to comprehend different option risks, benefits and consequences, and to make appropriate medical judgments at the time [3]. This collaborative process enables patients to make decisions about the proper care for themselves at the time.

Healthcare professionals encounter many healthcare conditions for which SDM can be helpful and applied, such as choice of labor after a previous Cesarean section [4], selection of dialysis modality [5], options for smoking cessation [6], location of end-of-life hospice [7], treatment for acute myeloid leukemia [8], treatment of de-escalation regimens in patients with RA on remission [9], management of valve heart disease [10], and breast reconstruction for patients with breast cancer [11]. Clinicians tend to make decisions based on medical knowledge and evidence, while patients are sensitive to the rigors of recovery and quality of life [12]. Patient-centered care should reflect treatment decisions that are consistent with patient preferences and goals [13]. A successful SDM talk ensures consistency in the personalized health model. It is likely to reduce decision regret and to increase satisfaction with the treatment [14]. A recent systematic review showed that SDM helps patients improve their understanding of treatment options, clarifies congruence between their values and medical choices, and engages them in decision making [15]. SDM is associated with increased patient knowledge, reduced anxiety about treatment, and diminished decision hesitance [8]. Despite the growing evidence supporting the importance of SDM and the decades of academic and policy calling for its adoption, SDM implementation remains suboptimal in clinical practice [16,17]. In general, clinicians are less able to promote SDM and require further training. Although clinicians have easy access to educational training on SDM, there is a lack of training guidelines on how to use it in clinical application [18,19]. There is also limited evidence for the intervention of training programs or the increased adoption of SDM by healthcare professionals [20].

The competency model is a set of success factors that include critical behaviors required to excel in a given role [21]. Further, competency models can be used to determine employees’ competencies in order to improve their performance in their current jobs or to prepare them for other jobs [22]. It is difficult for one person to master a wide range of abilities simultaneously. To effectively launch capability applications, it is beneficial to focus on a few critical capabilities and to implement them progressively [23]. However, most previous studies on the implementation of the SDM process have only focused on a single method or strategy, such as the SHARE (seek, help, assess, reach, evaluate) approach or the three-talk model [24,25], while overlooking the interaction between different affecting factors. Additionally, literacy is the key to improving peoples’ capabilities and to accomplishing many other rights [26]. In most clinical situations, clinicians are often the originators of medical decisions and should be acquainted with participating in the SDM process. Physicians’ higher awareness of the uncertainty about treatment options could improve their communication performance in decision making [27]. However, most healthcare professionals do not enter the clinical setting with adequate SDM skills and may be unfamiliar with the perception of SDM [28]. We may conceptualize “SDM” as a literacy or competency practice for physicians based on patient-centered care settings. Hence, the question arises of how to segment a set of competencies to facilitate physicians’ competency development to adequately perform SDM tasks. The decision-making trial and evaluation laboratory (DEMATEL) approach [29] is suitable to address this issue. The DEMATEL method facilitates the gathering of knowledge to form structural models and to visualize the causal relationships of subsystems through cause-and-effect diagrams.

The government of Taiwan officially launched the promotion of SDM strategies in 2016 [30]. SDM is a new trend for Taiwan’s medical society and public. Even though a few studies about concerns during SDM and training programs for healthcare providers have been conducted [31,32], there is currently a limited literature discussing the core competence of SDM tasks among clinicians in Taiwan. Therefore, this study intends to explore the priority and causal relationship of physician competency in implementing SDM tasks by using an online questionnaire.

The presented study has been organized into seven sections. The research background of SDM competency issues is explained in Part 1. The critical driving forces of SDM competency development are explored in Part 2. The proposed framework for the research is discussed in Part 3. The methodology to analyze the IAA-NRM (importance awareness analysis and network relation map) approach has been detailed in Part 4. Data analysis and related results have been presented in Part 5. Discussions of the research findings with diverse stakeholders’ perspectives on the core competence of SDM tasks are provided in Part 6. Finally, Part 7 concludes with the value of this model to assist hospital administrators and directors of medical education in recommending strategic directions for SDM competency development and talent cultivation programs.

## 2. Materials and Methods

The flow chart of this research work to assess the development of physicians’ ability to perform SDM tasks is shown in Figure 1.

The eight steps in the proposed model are as follows.

**Step** **1:**Define the critical decision problems for performing SDM tasks. **Step** **2:**Identify the driving aspects/criteria for developing SDM execution capability through a review of the literature and expert interviews. **Step** **3:**Investigate the level of importance and awareness of each aspect/criterion and evaluate the state of importance and awareness using the IAA method. **Step** **4:**Construct the cause–effect influence relation structure for SDM competency development via the NRM analysis.**Step** **5:**Integrating the IIA and NRM. The IIA defines aspects that are important and unaware of SDM competency. Meanwhile, the NRM traces the key aspects that handle SDM development. Then, establish the adopted strategy by merging the findings of the IAA and NRM approach.**Step** **6:**Combine the adoption paths of the importance indicator and awareness indicator via the rank of aspects and determine the suited adoption paths.**Step** **7:**Identify common adoption paths using ranking the aspects for different physician stakeholders.**Step** **8:**Select the paths that have both suited adoption paths and common adoption paths as common suited adoption paths through the IAA-NRM method.

This study used Microsoft Office Excel version 2016 (Microsoft, Redmond, WA, USA) to calculate the IAA approach and Matlab version R2017b software (The Mathworks Inc., Natick, MA, USA) to determine the NRM approach.

### 2.1. Establish the Content of the SDM Competencies

This study was based on various physicians’ viewpoints of SDM. We constructed the concept, content, and indicators of physicians’ professional competence in SDM tasks through the literature research and expert talks. We conducted expert interviews to obtain the following aspects of SDM competence. Four experienced experts in the Shared Decision-Making Group at the Centre for Quality Management, Kaohsiung Chang Gung Memorial Hospital were involved in this study. Several potential aspects/criteria were selected based on the literature collection and analysis, the results of which were used as the main interview question. After combining interviews with experts and the literature research, the information was organized into the source of a questionnaire design based on the subsequent definitions.

#### 2.1.1. Perception Assessment and Practice (PP)

SDM sought to ameliorate information asymmetry between health providers and patients. The initiation of the SDM process is indicated for a patient with an uncertain prognosis or severe illness, patients’ values and preferences vary wildly, or who encounters difficulty choosing between options [12,33]. The physician needs to orient patients to the SDM process and explain the importance of SDM to the patient [34]. Physicians’ inability to correctly identify the presence of SDM, to answer inquiries about SDM content, or to inadvertently neglect to enforce SDM are the barriers to implementing SDM [35]. A model of SDM presented by Hoffmann et al., in 2004 reviewed evidence-based medicine and the practice of patient-centered communication to support SDM within consultations [36]. The evidence-based approach combines individual clinical expertise with the best available external evidence. The physician could agree with specific elements of SDM supported by evidence-based medicine [37]. In short, physicians’ SDM perceptions are one of the core competencies necessary to promote SDM. Associated with the aspect of PP (perception assessment and practice), there are four evaluation criteria: concept and connotation (PP1), values and identity (PP2), perception and assessment (PP3), knowledge of evidence-based medicine (PP4), as shown in Table 1.

#### 2.1.2. Execution Process and Skills (ES)

How to carry out the steps of SDM: Previous studies have evaluated SDM processes. For example, the SHARE approach is a five-step process that involves analyzing and comparing each option’s benefits, harms, and risks through a meaningful conversation about what is most important to the patient [24]. The three-talk model is a three-stage process that involves choice talk, option talk, and decision talk [25]. Patient decision aid allows for matching patients’ values to their choices, reduces decisional conflict, helps the undecided to decide, improves understanding of options and outcomes, and guides patients to more pragmatic expectations [38]. Clinicians can encourage patients to use the questions and engage patients in value and self-efficacy elicitation and respect patient autonomy [34,39]. SDM lets physicians use evidence-based information while positioning the patient (and, when suitable, family members) at the center of clinical decisions. Training in SDM skills is part of evidence-based practice [37]. So, physicians should have the skills and execution needed to perform SDM. Concerning the ES (execution process and skills) aspect, there are four evaluation criteria: understand the decision process and steps (ES1), skills for evidence-based medicine (ES2), assistance of decision-making tools (ES3), and patient engagement and guidance (ES4), as shown in Table 1.

#### 2.1.3. Physician–Patient Relationship and Interaction (RI)

The ability of physicians to develop good relationships with patients is one of the crucial goals of healthcare and it is vital in undertaking SDM [40]. Effective communication is an important element of the doctor–patient relationship. Lack of a good doctor–patient relationship causes lower patient satisfaction, insufficient understanding of interventions, and poor compliance with treatments [8]. Communication skills are needed for patient-centered care [41]. Physicians can appreciate patients’ culture, education, and language diversity and can accept the health values, beliefs, views, and feelings of patients and their families [8,42]. There are many ways to improve communication skills, such as using simple language, responding to teaching, verbal and nonverbal methods, gathering adequate information, checking the patient’s understanding, and expressing empathy [42,43]. Clinicians should consider the SDM process across disciplines to permit the exchange of information, deliberation, and the joint decision making regarding essential treatments [44]. Interprofessional SDM is when healthcare providers work together to engage patients in decision making and to improve the quality of decision making by facilitating the integration of healthcare services [45]. Therefore, clinicians must communicate and interact with their patients and team members well during the SDM process. Associated with the RI (physician–patient relationship and interaction) aspect, there are four evaluation criteria: doctor–patient relationship building (RI1), verbal or nonverbal communication (RI2), teaching back and execute (RI3), and team coordination and cooperation (RI4), as shown in Table 1.

#### 2.1.4. Shared Information and Decision Making (SD)

Physicians must inform the patient that a decision needs to be made and explain the patient’s preferred role in decision making [46]. The health providers share evidence-based information regarding available options and discuss the benefits, risks, costs, and uncertainty of each, while allowing adequate time for questions [47]. Flexibility in the manner that physicians conduct the SDM process is essential [48]. Physicians understand patients’ values, concerns, and preferences and support patients in their deliberations. The parties discuss the decision, make or postpone the decision as appropriate, and arrange follow-ups [49]. Patients can revisit decisions if available therapy options do not deliver the desired health effects [12]. Thus, how to practice shared information and decision making would be one of the important physicians’ competencies for SDM. Concerning the SD (shared information and decision making), there are four evaluation criteria: decision needs assessment (SD1), information sharing (SD2), co-participation decision making (SD3), and decision tracking and evaluation (SD4), as shown in Table 1.

We conducted the literature reviews and expert interviews to determine the critical aspects/criteria, as shown in Table 1. There are four aspects, including perception assessment and practice (PP), execution process and skills (ES), physician–patient relationship and interaction (RI), and shared information and decision making (SD). Each aspect includes four criteria as described in Table 1.

**Table 1 healthcare-10-01844-t001:** The descriptions of aspects/criteria of SDM competency development for physicians.

Aspects/Criteria	Item Descriptions	References
Perception assessment and practice (PP)		
SDM concept and connotation (PP1)	Physicians can effectively understand the definition and connotation of SDM.	[1,35,36]
SDM values and identity (PP2)	Physicians can effectively understand the importance of SDM and agree with the value of SDM.	[8,34]
SDM perception and assessment (PP3)	Physicians can guide patients to make SDM from patients’ representations and contexts.	[12,33]
Knowledge of evidence-based medicine (PP4)	Emerging medical information and evidence-based medicine can help improve physicians’ ability to accomplish SDM.	[1,37]
Execution process and skills (ES)		
Understand decision process and steps (ES1)	Physicians can have the expertise required to implement SDM in medical consultation.	[1,24,25]
Skills for evidence-based medicine (ES2)	Physicians have skills for evidence-based medicine to assist the SDM process.	[36,37]
Assistance of decision-making tools (ES3)	Physicians can use digital media materials and patient decision aids to oblige medical decision making and enhance communication.	[15,38]
Patient engagement and guidance (ES4)	With respect for patient autonomy in decision making, physicians can advise patients to express personal opinions and encourage mutual participation promptly.	[34,39]
Physician–patient relationship and interaction (RI)		
Doctor–patient relationship building (RI1)	To face patients with different characteristics and backgrounds, physicians can fully understand, endure, and effectively establish an excellent doctor–patient relationship.	[8,40,41]
Verbal or nonverbal communication (RI2)	Physicians should communicate messages in a way that is easy for patients to comprehend and allow them to understand before expressing their wishes fully.	[8,42]
Teaching back and execute (RI3)	Physicians should take appropriate response teaching and confirm that the patient can understand and accurately execute the content of the response teaching.	[43]
Team coordination and cooperation (RI4)	Physicians should learn to work with medical teams and other healthcare staff to improve the quality and outcomes of patient care.	[44,45]
Shared information and decision making (SD)		
Decision needs’ assessment (SD1)	Physicians should understand patients’ value and preferences and apply evidence-based medical information to guide patients on diverse treatment options.	[1,46,50]
Information sharing (SD2)	Physicians should share medical information with patients, explain the benefits and risks of different medical treatments, and further clarify their concerns and doubts, so patients can select the most appropriate therapy from various options.	[47,48,50]
Co-participation decision making (SD3)	Physicians can allow patients and their families to participate in decision making. While patients fully understand the treatment plan with proper doctor–patient contact, doctors make final determinations with patients and document the reasons and content of the decision.	[37,49]
Decision tracking and evaluation (SD4)	Physicians can follow up after the medical decision making and evaluate the treatment effect and patient satisfaction with the decision making.	[12,49]

### 2.2. Questionnaire Design and Reliability Analysis

The study was conducted in accordance with the principles of the Declaration of Helsinki and approved by the Institutional Review Board of the Chang Gung Memorial Hospital (IRB No: 202200716B0, 202200716B0C501). We analyzed stakeholders on the aspects/criteria of the potential questionnaire and then developed the stakeholders’ survey. The study utilized an 11-point Likert Scale (0~10) to gather stakeholders’ views on the importance and awareness of the driving aspects/criteria. Data were collected through online questionnaires.

### 2.3. The IAA Approach

The IIA approach was according to the importance-performance analysis that Martilla and James [51] proposed, but instead of “performance”, “awareness” is used in this study. Several studies demonstrated an evaluation model using a similar IAA method to explore the importance and relationship between factors [52,53,54,55]. For example, Tonge et al. [52] used importance satisfaction analysis (ISA) to evaluate the service quality gap in tourism research. Wang et al. [53] developed a proposed model to integrate the satisfaction importance analysis (SIA) for evaluating the performance of each section and the DEMATEL technique for capturing causal effects to develop an impact-relations map. Lin, C., Ref. [55], combined the innovation opportunity analysis–network relationship mapping (IOA-NRM) method to determine the appropriate value-driven tourism transformation strategy. The study estimates each aspect’s importance and awareness status and normalizes these collected data. In the normalized procedure, these aspects can be separated into four quadrants: (1) the first quadrant represents the high importance and high awareness level (H, H), (2) the second quadrant represents the low importance and high awareness level (L, H), (3) the third quadrant represents the low importance and low awareness level (L, L), and (4) the fourth quadrant represents the high importance and low awareness level (H, L).

### 2.4. The DEMATEL Approach

Many studies have used DEMATEL methods through the NRM approach to solve complex decision issues, such as the analysis of user interface [56], the airline safety management system [57], building the value-created system in the science (technology) park [58], identifying product positions by the hybrid multiple criteria decision-making (MCDM) model [59], selection of driving aspects for digital music service platforms by a hybrid MCDM approach [60], the structural model of sustainable consumption and production adoption based on a grey-DEMATEL approach [61], evaluating the model of culture festival events’ service by the MCDM approach [62], the analysis of the med-tech industry entry strategy during the COVID-19 pandemic [63], identification of the key success factors of SDM [12], determining urban revitalization and regional development strategies considering urban stakeholders [64], exploring the hindering factors for medical information standards’ dissemination [65], defining the critical factors for urban music festival tourism [66], identifying key performance indexes of hospital management by double hierarchy hesitant fuzzy linguistic term sets (DHHFL)–DEMATEL method [67], exploring the relationship among acute kidney injury risk factors, with or without COVID-19 [68], adopting the fuzzy DEMATEL method to determine the driving forces of enterprise environmental responsibility for the Chinese auto manufacturing industry [69], and analyzing the critical factors affecting SDM from orthopedic nurses’ perspectives [70]. 

The DEMATEL method consists of five steps: (1) estimate the original average matrix; (2) compute the direct influence matrix; (3) compute the indirect influence matrix; (4) evaluate the full influence matrix; (5) examine the NRM relationship.

(1).Estimate the original average matrix

Respondents rated the impact of each aspect on the others on a scale of 4 to 0; “4” indicates the extreme impact on others; “0” denotes no impact on others between aspects/criteria; “3”, “2”, and “1” represent “high influence”, “medium influence”, and “low influence on others”. 

(2).Compute the direct influence matrix 

From Equations (1) and (2), the *D* (direct influence matrix) can be obtained from *A* (initial average influence matrix). The *D* (direct influence matrix) denotes each direct influence value, and the numbers are 0 on the diagonal. We calculate the sum of each row and column to get the direct effect value. The sum of each column and row is at most 1 (only one is equal to 1) in the matrix.
(1)D=sA,  s>0
where
(2)$$s=\underset{i,j}{\min}\;[1/\underset{1\leq i\leq n}{\max}\sum_{j=1}^{n}a_{ij},1/\underset{1\leq j\leq n}{\max}\sum_{i=1}^{n}a_{ij}],i,j=1,2,\ldots ,n$$
and $\underset{m\to \infty }{\lim}\;D^{m}={\left[0\right]}_{n\times n}$, where $D={\left[x_{ij}\right]}_{n\times n}$, when $0<\sum_{j=1}^{n}x_{ij}\leq 1\;$ or $0<\sum_{i=1}^{n}x_{ij}\leq 1$, and at least one $\sum_{j=1}^{n}x_{ij}$ or $\sum_{i=1}^{n}x_{ij}$ equals one, but not all. Thus, we can guarantee $\underset{m\to \infty }{\lim}\;D^{m}={\left[0\right]}_{n\times n}$.

(3).Compute indirect influence matrix

The indirect influence matrix can be obtained from Equation (3).
(3)$$ID=\sum_{i=2}^{\infty }D^{i}=D^{2}{\left(I-D\right)}^{-1}$$

(4).Calculate the full influence matrix

*T* (full influence matrix) can be obtained by Equation (4) or (5). As illustrated in Equation (6), *T* (total influence matrix) has numerous elements. The ***d*** in Equation (7) is the sum of row values, ***r*** in Equation (8) is the sum of column values, $d_{i}+r_{i}$ is the sum of row values plus column values, and $d_{i}-r_{i}$ is the sum of row values minus column values. If $d_{i}-r_{i}$ > 0, this aspect has a more significant influence on other aspects. If $d_{i}-r_{i}$ < 0, this aspect is more influenced by other aspects.
(4)$$T=D+\mathrm{ID}=\sum_{i=1}^{\infty }D^{i}$$
(5)$$T=\sum_{i=1}^{\infty }D^{i}=D{\left(I-D\right)}^{-1}$$
(6)$$T=[t_{ij}],\;i,j\in \{1,2,\ldots ,n\}$$
(7)$$d=d_{n\times 1}=[\sum_{j=1}^{n}t_{ij}{]}_{n\times 1}=(d_{1},\ldots ,d_{i},\ldots ,d_{n})$$
(8)$$r=r_{n\times 1}=[\sum_{i=1}^{n}t_{ij}{{]}^{\prime }}_{1\times n}=(r_{1},\ldots ,r_{j},\ldots ,r_{n})$$

(5).Examine the NRM (network relation map)

From Equation (9), the net full influence matrix $C_{net}$ is determined. The diagonal entries in the matrix are all 0. Put simply, the matrix includes a precisely lower triangular matrix and a purely upper triangular matrix. In addition, upper and lower triangular matrices have the same values but opposite signs. This property allows choosing one of the purely triangular matrices. The net influence matrix is applied, obtained by Equation (9).
(9)$$C_{net}=[t_{ij}-t_{ji}],\begin{array}{cc} & i,j\in \{1,2,\ldots ,n\} \end{array}$$

### 2.5. The IAA-NRM Approach

The analytic process of IAA-NRM (importance awareness analysis–network relation map) contains two phases. The first phase is the IAA approach and the second phase is the NRM approach. The IAA analysis delimits the importance and awareness level of aspects/criteria statuses for SDM competencies; the IAA analysis can help leaders of physicians’ training determine criteria that should be developed when the standard importance level is less than the average importance degree.

## 3. Results

### 3.1. Reliability Analysis

A total of 146 physicians’ questionnaires were gathered, and 124 were valid samples (34 attending physicians, 35 medical residents, and 55 doctors in post-graduate years (PGYs)). Cronbach’s alpha verified the reliability of importance and awareness. The reliability of importance and awareness indicators are 0.969 and 0.964. Both are higher than 0.7, meaning that the importance and awareness indicators have high reliability. The reliability of the competency aspect is 0.947, which indicates that the reliability of this model is highly consistent (Table 2).

### 3.2. The IAA-NRM Approach

Table 3 presents the evaluation obtained using IAA and NRM. The IAA analysis of this study is presented below: The first adoption step is to improve those aspects (i.e., PP, ES) falling into the third quadrant (L, L), indicating the low importance degree (SI < 0) and low awareness degree (AI < 0). The second adoption step is to enhance those aspects (i.e., RI, SD) falling into the first quadrant (H, H), showing the high importance degree and high awareness degree, as shown in Figure 2, left, Appendix A. Moreover, NRM analysis reveals that the PP and ES factors are in the cause group, while the other two aspects, SD and RI, are in the effect group (Figure 2, right and Appendix A). The PP aspect influences the aspects of ES, SD, and RI, and the ES aspect affects the aspects of SD and RI. Then, the SD aspect affects the RI aspect. Therefore, the best strategy for SDM competency development is to improve PP. Overall, adoption strategy A (keeping strategy, which requires maintenance) can apply to the aspects of SD and RI aspects. Adoption strategy C (focus status, which requires strengthening) can apply to the aspects of PP and ES, as shown in Figure 2 and Table 3.

### 3.3. Evaluation of the Suited Adoption Paths via the Rank of Aspects

Through the suited adoption path analysis, the II (importance indicator) ranking is RI > SD > PP > ES, and the AI (awareness indicator) ranking is SD > RI > ES > PP. The four available paths (PP→RI; PP→SD→RI; PP→ES→RI; PP→ES→SD→RI) can be identified using the NRM approach. The advantage aspects can improve the disadvantage aspects. The underlined paths indicate the effective paths. The II ranking is RI > SD > PP > ES. The PP aspect can impact the aspect of ES by the third available path (PP(3)→ES(4)→RI(1)). The PP aspect can influence the ES aspect by the fourth available path (PP(3)→ES(4)→SD(2)→RI(1)), as shown in Table 4. The AI ranking is SD > RI > ES > PP, and the SD aspect can influence the RI aspect through the second available path (PP(4)→SD(1)→RI(2)). The SD aspect can influence the RI aspect through the fourth available path (PP(4)→ES(3)→SD(1)→RI(2)). The adoption paths of the importance indicator and awareness indicator were integrated and found one suitable adoption path (PP→ES→SD→RI), as shown in Table 4.

### 3.4. Determination of Common Adoption Paths Using the Aspects Rank for Different Stakeholders

In the attending physicians’ opinions, the common adoption path analysis of the II (importance indicator), the II ranking is RI > PP > SD > ES. In the medical residents’ opinions, the II ranking is SD > RI > PP >ES. This study can find the available paths (PP→RI; PP→SD→RI; PP→ES→RI; PP→ES→SD→RI) through the NRM approach, and then the advantages’ aspect can improve the disadvantages’ aspect. According to the attending physicians’ opinions, the ranking of the II is RI > PP > SD > ES. The aspect of PP can influence the SD aspect by the second available path (PP(2)→SD(3)→RI(1)). The PP aspect can influence the ES aspect by the third available path (PP(2)→ES(4)→RI(1)). The PP aspect can influence the ES aspect by the fourth available path (PP(2)→ES(4)→SD(3)→RI(1)), as shown in Table 5. According to the medical residents’ opinions, the ranking of the II is SD > RI > PP >ES. The aspect of SD can influence the RI aspect by the second available path (PP(3)→SD(1)→RI(2)). The PP aspect can influence the ES aspect by the third available path (PP(3)→ES(4)→RI(1)). The PP aspect can influence the ES aspect and the SD can impact the RI aspect through the fourth available path (PP(3)→ES(4)→SD(1)→RI(2)), as shown in Table 5.

In the common adoption path analysis of AI (awareness indicator), according to the attending physicians’ opinions the AI ranking is SD > RI > PP = ES, and the AI ranking is SD > RI > PP > ES based on the medical residents’ opinions. This study can obtain the four available paths (PP→RI; PP→SD→RI; PP→ES→RI; PP→ES→SD→RI) using the NRM approach. According to the attending physicians’ opinions, the ranking of the AI is SD > RI > PP = ES. The aspect of SD can influence the RI aspect by the second available path (PP(3)→SD(1)→RI(2)). The PP aspect can influence the ES aspect by the third available path (PP(3)→ES(3)→RI(2)). The PP aspect can influence the ES aspect and the SD can influence the RI aspect in the fourth available path (PP(3)→ES(3)→SD(1)→RI(2)), as shown in Table 5. According to the medical residents’ opinions, the AI ranking is SD > RI > PP > ES. The aspect of SD can influence the aspect of RI in the second available path (PP(3)→SD(1)→RI(2)). The PP aspect can influence the ES aspect based on the third available path (PP(3)→ES(4)→RI(2)). The PP aspect can influence the ES aspect and the SD can enhance the RI aspect by the fourth available path (PP(3)→ES(4)→SD(1)→RI(2)), as shown in Table 5.

### 3.5. Explore the Available Paths and Suitable Adoption Paths for Various Levels of Physicians to Perform SDM Tasks

#### 3.5.1. The Suited Adoption Paths for Attending Physicians

The adoption strategies and suited adoption paths were presented by the IAA-NRM approach in the attending physicians’ opinions. The RI aspect is located in the first quadrant (H, H), displaying that the RI aspect is characterized by a high II and a high AI. The attending physicians should continue to maintain RI capacity. The ES aspect is located in the third quadrant (L, L), showing that the ES aspect is defined by a low II and a low AI. Although attending physicians do not become aware of the ES aspect, attending physicians would focus on improving the ES aspect through aggressive intervention.

The PP and ES aspects have a positive net influence effect (*d* − *r* > 0), so the PP aspect can influence the ES aspect. The PP aspect is only enhanced through itself. Four adoption strategies are presented in Table 6. Adoption strategy A (keeping strategy) can apply to the RI aspect, so the RI aspect should be kept continually. Adoption strategy B (attention strategy) can apply to the SD aspect. Adoption strategy C (focus strategy) can apply to the ES aspect, so the ES aspect should be strengthened. Adoption strategy D (monitoring strategy) can apply to the PP aspect. The RI aspect is located in the first quadrant (H, H), so the RI aspect should be kept continually. The RI aspect can be influenced by the PP, ES, and SD aspects, as indicated in Figure 3 and Table 6.

In the analysis of suitable adoption paths, the II (importance indicator) ranking is RI > PP > SD > ES, and the AI (awareness indicator) ranking is SD > RI > PP = ES for the attending physicians. The NRM approach encounters the four available paths (PP→RI; PP→SD→RI; PP→ES→RI; PP→ES→SD→RI). The II (importance indicator) ranking is RI > PP > SD > ES. The PP aspect can influence the SD aspect via the second available path (PP(2)→SD(3)→RI(1)). The aspect of PP can influence the ES aspect via the third available path (PP(2)→ES(4)→RI(1)). The PP aspect can improve the ES aspect via the fourth available path (PP(2)→ES(4)→SD(3)→RI(1)). The AI ranking is SD > RI > PP = ES. The SD aspect can influence the RI aspect via the second available path (PP(3)→SD(1)→RI(2)). The aspect of PP can influence the ES aspect via the third available path (PP(3)→ES(3)→RI(1)). The aspect of PP can influence the ES aspect and the SD aspect can improve the RI aspect via the fourth available path (PP(3)→ES(3)→SD(1)→RI(2)). Additionally, the IAA-NRM method combines the available paths of the II (importance indicator) and the AI (awareness indicator). There are three suitable adoption paths (PP→SD→RI; PP→ES→RI; PP→ES→SD→RI) for the attending physicians, as indicated in Table 7.

#### 3.5.2. The Suited Adoption Paths for Medical Residents

According to the medical residents’ opinions, the adoption strategies and paths were present via the IAA-NRM approach. The SD and RI aspects are located in the first quadrant (L, L), indicating that the SD and RI aspects were characterized by a high II and a high AI. Medical residents should continue to maintain RI and SD capacity. The PP and ES aspects are located in the third quadrant (L, L), showing that the PP and ES aspects were defined by a low II and a low AI. Although medical residents do not become aware of the PP and ES aspects, medical residents should take the ability to improve the state of both seriously.

The PP and ES aspects have a positive net influence effect (*d* − *r* > 0) by the NRM approach, thus the aspect of PP can influence the ES aspect. The PP aspect only improves through itself. There are four adoption strategies proposed in Table 8. The adoption strategy A (keeping strategy) can apply to the RI and SD aspects, so the two aspects should be kept continuously. Adopting strategy C (focus strategy) can apply to the PP and ES aspects, so the two aspects should be focused to improve intensely. We can determine that the aspects of RI and SD should be kept, and the PP is the aspect that is the primary aspect with net influence, such that the PP aspect can influence the ES aspect. We can enhance the RI aspect by handling the PP, ES, and SD aspects. The RI aspect is the primary aspect being affected; hence, the aspect of RI can be improved by the aspects of PP, ES, and SD, as illustrated in Figure 4 and Table 8.

In the analysis of the suitable adoption paths, the II (importance indicator) ranking is SD > RI > PP > ES, and the AI (awareness indicator) ranking is SD > RI > PP > ES for the medical residents. Four available paths (PP→RI; PP→SD→RI; PP→ES→RI; PP→ ES→SD→RI) are demonstrated by the NRM approach. The II (importance indicator) ranking is SD > RI > PP > ES. The SD aspect can influence the RI aspect via the second available path (PP(3)→SD(1)→RI(2)). The aspect of PP can influence the ES aspect via the third available path (PP(3)→ES(4)→RI(1)). The PP aspect can improve the ES aspect and the SD aspect can improve the RI aspect via the fourth available path (PP(3)→ES(4)→SD(1)→RI(2)). The AI (awareness indicator) ranking is SD > RI > PP > ES. The SD aspect can influence the RI aspect via the second available path (PP(3)→SD(1)→RI(2)). The aspect of PP can influence the ES aspect via the third available path (PP(3)→ES(4)→RI(1)). The PP aspect can improve the ES aspect and the SD aspect can improve the RI aspect via the fourth available path (PP(3)→ES(4)→SD(1)→RI(2)). Additionally, the IAA-NRM approach combines the available paths of the II (importance indicator) and AI (awareness indicator). There are three suitable adoption paths (PP→SD→RI; PP→ES→RI; PP→ES→SD→RI) for medical residents, as shown in Table 9.

#### 3.5.3. The Suited Adoption Paths for PGYs

The adoption strategies and adoption paths were present via the IAA-NRM approach in the PGYs’ opinions. The RI and SD aspects were located in the first quadrant (H, H), indicating that the RI and SD aspects were characterized by a high II and a high AI. The PGYs should keep the RI and SD aspects firmly in mind. The PP and ES aspects were located in the third quadrant (L, L), showing that the PP and ES aspects were defined by a low II and a low AI. Although PGYs were not aware of the PP and ES, the PGYs should adopt intense improvement in the status. The PP and ES aspects have a positive net influence effect (*d* − *r* > 0) by the NRM approach, so that the PP aspect can influence the ES aspect. The PP aspect only improves through itself. Four adoption strategies are presented in Table 10. Adoption strategy A (keeping strategy) can apply to the RI and SD aspects, and adoption strategy C (focus strategy) can apply to the PP and ES aspects. We can establish that the RI and SD aspects should be kept, and the PP is the primary aspect with positive net influence, so the PP aspect can influence the ES aspect. We can improve the RI aspect by managing the PP, ES, and SD aspects. The RI aspect is the primary aspect being influenced; thus, the RI aspect can be improved by the aspects of PP, ES, and SD, as shown in Figure 5 and Table 10.

In the analysis of the suitable adoption paths, the II (importance indicator) ranking is RI > SD > ES > PP, and the AI (awareness Indicator) ranking is SD > RI > ES > PP for the PGYs. Four available paths (PP→RI; PP→SD→RI; PP→ES→RI; PP→ES→SD→RI) are observed by the NRM approach. The II ranking is RI > SD > ES > PP. There are no available paths in the II (importance indicator). The AI ranking is SD > RI > ES > PP, the SD aspect can improve the RI aspect in the second available path (PP(4)→SD(1)→RI(2)). The SD aspect can influence the RI aspect by the fourth path (PP(4)→ES(3)→SD(1)→RI(2)). In addition, the IAA-NRM approach integrated the available paths of the II (importance indicator) and AI (awareness indicator), but there were no suited adoption paths for PGYs, as shown in Table 11.

### 3.6. Explore Common Adoption Paths for Various Levels of Physicians to Perform SDM Tasks

#### 3.6.1. Common Adoption Paths for II (Importance Indicator)

In the attending physicians’ opinions, the ranking of the II (importance indicator) is RI > PP > SD > ES. According to the opinion of medical residents, the II ranking is SD > RI > PP > ES. The II ranking is RI > SD > ES > PP in the PGYs’ opinion. Four available paths are observed by the NRM approach (PP→RI; PP→SD→RI; PP→ES→RI; PP→ES→SD→RI). In the attending physicians’ opinions, the II order is RI > PP > SD > ES. The PP aspect can improve the SD aspect by the second available path (PP(2)→SD(3)→RI(1)). The PP aspect can influence the ES aspect by the third available path (PP(2)→ES(4)→RI(1)). The PP aspect can influence the ES aspect by the fourth available path (PP(2)→ES(4)→SD(3)→RI(1)). The II order is RI > SD > ES > PP, according to the medical residents’ opinions. The SD aspect can influence the RI aspect by the second available path (PP(3)→SD(1)→RI(2)). The PP aspect can influence the ES aspect by the third available path (PP(3)→ES(4)→RI(2)). The PP aspect can influence the ES aspect, and the SD can influence the RI aspect by the fourth available path (PP(3)→ES(4)→SD(1)→RI(2)), as presented in Table 12. The II (importance indicator) order is RI > SD > ES > PP in the PGYs’ opinions. There are no available paths in the II (importance indicator) for the PGYs’ opinions, as presented in Table 12; hence, the IAA-NRM method incorporates the various viewpoints of attending physicians, medical residents, and PGYs. No common adoption path was found, as shown in Table 12.

#### 3.6.2. Common Adoption Paths for AI (Awareness Indicator)

In the analysis of common adoption paths, in the attending physicians’ opinions the AI (awareness indicator) ranking is SD > RI > PP = ES, and the AI ranking is SD > RI > PP > ES according to the medical residents’ opinions. The AI ranking is SD > RI > ES > PP in the PGYs’ opinions. Four available paths were obtained by the NRM approach (PP→RI; PP→SD→RI; PP→ES→RI; PP→ES→SD→RI). The AI ranking is SD > RI > PP = ES in the opinions of attending physicians. The SD aspect can influence the RI aspect by the second path (PP(3)→SD(1)→RI(2)). The PP aspect can influence the ES aspect by the third available path (PP(3)→ES(3)→RI(2)). The PP aspect can influence the ES aspect and the SD aspect can influence the RI aspect by the fourth available path (PP(3)→ES(3)→SD(1)→RI(2)). The AI ranking is SD > RI > PP > ES in the opinions of medical residents. The SD aspect can influence the RI aspect by the second available path (PP(3)→SD(1)→RI(2)). The PP aspect can influence the ES aspect via the third available path (PP(3)→ES(4)→RI(2)). The PP aspect can influence the ES aspect and the SD can influence the RI aspect by the fourth path (PP(3)→ES(4)→SD(1)→RI(2)), as shown in Table 13. The AI ranking is SD > RI > ES > PP in the opinions of PGYs. The SD aspect can influence the RI aspect by the second path (PP(4)→SD(1)→RI(2)). The SD aspect can influence the RI aspect by the fourth available path (PP(4)→ES(3)→SD(1)→RI(2)). Hence, the IAA-NRM approach combines the viewpoints of attending physicians, medical residents, and PGYs. There are two common adoption paths (PP→SD→RI; PP→ES→SD→RI), as shown in Table 13.

## 4. Discussion

### 4.1. Comparison of Suitable Adoption Paths among Different Physician Stakeholders

According to the attending physicians’ opinions, the RI aspect was in the first quadrant and the SD aspect was in the second quadrant; the ES aspect was in the third quadrant and the PP aspect was in the fourth quadrant. Adoption strategy A (keeping strategy) can operate the RI aspect and adoption strategy B (attention strategy) can adapt to the SD aspect. Adoption strategy C (focus strategy) can be used in the ES aspect. Adoption strategy D (monitoring strategy) can apply to the PP aspect for attending physicians. Three suitable adoption paths for attending physicians were observed (PP→SD→RI; PP→ES→RI; PP→ES→SD→RI). The second suitable adoption path is that the PP aspect influences the SD aspect, which impacts the RI aspect. The third suitable adoption path is that the PP aspect influences the ES aspect, which impacts the RI aspect. The fourth suitable adoption path is that the PP aspect influences the ES aspect. The ES aspect influences the SD aspect and the SD aspect influences the RI aspect, as shown in Table 14.

The medical residents’ opinions indicate that the SD and RI aspects are in the first quadrant. The PP and ES aspects are in the third quadrant. Adoption strategy A (keeping strategy) can use the SD and RI aspects, and adoption strategy C (focus strategy) can apply to the PP and ES aspects for medical residents. Three suitable adoption paths for medical residents are noted (PP→SD→RI; PP→ES→RI; PP→ES→SD→RI). The second suitable adoption path is that the PP aspect influences the SD aspect, which impacts the RI aspect. The third suitable adoption path is that the PP aspect influences the ES aspect, which impacts the RI aspect. The fourth suitable adoption path is that the PP aspect influences the ES aspect. The ES aspect influences the SD aspect and the SD aspect influences the RI aspect, as shown in Table 14. According to the opinions of PGYs, the SD and RI aspects are in the first quadrant and the aspects of PP and ES were in the third quadrant. Adoption strategy A (keeping strategy) can apply to the SD and RI aspects, and adoption strategy C (focus strategy) can adapt to the PP and ES aspects for PGYs. However, there is no suited adoption path for PGYs, as shown in Table 14.

### 4.2. Comparison of Common Adoption Paths among Different Physician Stakeholders

#### 4.2.1. Common Adoption Path of II (Importance Indicator)

In the opinion of senior physicians (attending physicians), the II ranking is RI > PP > SD > ES. Three adoption paths are presented (PP→SD→RI; PP→ES→RI; PP→ES→SD→RI). In senior physicians’ (medical residents) opinions, the II ranking is SD > RI > PP > ES. Three adoption paths were also demonstrated (PP→SD→RI; PP→ES→RI; PP→ES→SD→RI). According to the junior physicians (PGYs) opinions, the ranking of the II (importance indicator) is RI > SD > ES > PP. There is no adoption path available. This study combined the opinions of attending physicians, medical residents, and PGYs, and there are no common adoption paths, as shown in Table 14. However, there are three common adoption paths (PP→SD→RI; PP→ES→RI; PP→ES→SD→RI) according to the senior physicians’ (attending physicians and medical residents) opinions.

#### 4.2.2. Common Adoption Path of AI (Awareness Indicator)

In the opinion of the senior physicians (attending physicians), the AI ranking is SD > RI > PP = ES. In the opinion of the senior physicians (medical residents), the AI ranking is SD > RI > PP > ES. There are three common adoption paths (PP→SD→RI; PP→ES→RI; PP→ES→SD→RI) for the senior physicians (attending physicians and medical residents). In the opinion of the junior physicians (PGYs), the AI ranking is SD > RI > ES > PP. There are two adoption paths (PP→SD→RI; PP→ES→SD→RI) for the junior physicians (PGYs). The current study merges the opinions of senior physicians (attending physicians and medical residents) and junior physicians (PGYs). There are two common adoption paths (PP→SD→RI; PP→ES→SD→RI) in the AI. The second common adoption path is that the PP aspect influences the SD aspect, and the SD aspect influences the RI aspect. The fourth common adoption path is that the PP aspect influences the ES aspect, the ES aspect influences the SD aspect, and the SD aspect influences the RI aspect, as shown in Table 14.

### 4.3. Identify Common Suitable Adoption Paths among Different Physician Stakeholders

This study investigates various physicians, which include senior physicians (attending physicians, and medical residents) and junior physicians (PGYs). There is no common suitable path among the full physicians (attending physicians, medical residents, and PGYs), as shown in Table 14. However, there are three common suitable paths (PP→SD→RI; PP→ES→RI; PP→ES→SD→RI) for senior physicians (attending physicians and medical residents). The second common suitable path is that the PP influences the SD aspect and the SD aspect influences the RI aspect for senior physicians. The third common suitable path is that the PP influences the ES aspect and the ES aspect influences the RI aspect. The fourth common suitable path is that the PP influences the ES aspect and the ES aspect influences the SD aspect. The SD aspect influences the RI aspect, as indicated in Table 14. The senior physicians (attending physicians and medical residents) are the core roles of SDM performance in the hospital. Senior physicians’ higher-value viewpoints can enhance their SDM skills, enable them to share adequate information, make the right decision, and improve physician–patient relationships.

### 4.4. Review of Research Findings

The current study can help hospital administrators and directors of medical education re-examine the future direction of SDM implementation through various kinds of physicians and identify the adoption strategies and common suited paths for physicians’ training education in SDM. There are several findings to this study, which are as follows.

(1)Similarities in the IAA findings for all physicians: As shown in Table 4, in the II (importance indicator), the RI aspect is more crucial than the SD aspect, and the SD aspect is also more critical than the PP and ES aspects. Furthermore, the SD aspect is more awakened than the RI aspect, and the RI aspect is also more awakened than the ES and PP aspects in the AI (awareness indicator). Thus, hospital administrators and directors of medical education should learn that physicians focus more on SD and RI aspects (interactive process) than the PP and ES aspects (basic knowledge and skills) in the daily practice of SDM. However, the NRM approach revealed that the SD and RI aspects are influenced. So, hospital administrators and directors of medical education should keep sharing information, deepen decision making, and build effective physician–patient communication.(2)Differences in adoption strategies among different physician stakeholders: The PP and ES aspects are in the third quadrant (low importance and low awareness) for full sample. The PP and ES aspects are also in the third quadrant for medical residents and PGYs. However, the PP aspect shifts to the fourth quadrant (high importance and low awareness) for attending physicians. The SD and RI aspects are in the first quadrant (high importance and high awareness) for the full sample. The SD and RI aspects are also in the first quadrant (high importance and high awareness) for medical residents and PGYs. However, the SD aspect shifts to the second quadrant (low importance and high awareness) for attending physicians. Although all physicians (full sample) ignored the PP aspect, the PP aspect stands as the leading influencer in NRM analysis. As the goal of strategy C is to focus on strengthening abilities, we should discover paths to improve SDM perception and assessment, such as by the intervention of SDM training curriculum or standardized patients. For the residency or PGY training programs, SDM perception and assessment could be enhanced in the Objective Structured Clinical Examination (OSCE), Entrustable Professional Activity (EPA) evaluation, or simulation learning. (3)Focus on the key aspects driving SDM competency development: Our study observed that the PP aspect is the primary influencer of SDM competence development for all physician stakeholders by NRM analysis. Knowledge translation is defined as various stakeholders such as clinicians, patients, managers, and policy makers using knowledge in practice and decision making [71]. Knowledge and awareness among healthcare providers and patients, as well as decision aids and skills’ training, are needed for SDM to be more widely executed [49]. In a clinician’s opinion, common barriers to implementing SDM in clinical practice include lack of knowledge and familiarity with SDM, poor interpersonal skills, and time pressure [3,72]. Knowledge of SDM can increase healthcare professionals’ positive attitude and willingness to practice SDM [71]. The initiation of an SDM program requires consideration of the health provider’s knowledge and beliefs regarding SDM [4]. A previous study in Taiwan found that adequate knowledge of SDM among medical staff is one of the three most common major facilitators [30]. A recent systematic review found that there is a controversial effect to improving SDM knowledge and skills through the training programs targeting medical doctors [3]. A study for junior doctors in Denmark demonstrated that most of the survey respondents were satisfied with their SDM learning outcomes from the training course: knowledge (73%), competencies (57%), and communication skills (66%) [73]. The current study disclosed that physicians’ SDM perceptions of the background and rationale of SDM is the critical driving factor to promote SDM development. The IAA-NRM approach helps clinicians address the gaps of needs, find the best path, and inform the SDM development strategies. Therefore, this study suggests that the effective implementation of SDM perception can deepen decision making and facilitate the physician–patient relationship.(4)The SDM competency development for attending physicians: Attending physicians are required to address a wide range of issues, and they must master both clinical and many non-clinical tasks [74]. Moreover, as the initiators of medical decision making in clinical practice, the attending physicians should be familiar with and be able to participate in the SDM process. They appear to play an essential role in encouraging patient engagement and offering options [75]. SDM is an application that requires mapping competencies to specific clinical tasks in medical disciplines and considering medical needs and patient values and preferences. The SD aspect is in the second quadrant (attention strategy), the ES aspect is in the third quadrant (focus strategy), and the PP aspect is in the fourth quadrant (monitoring strategy) for attending physicians’ points of view. It seems that the attending physicians are familiar with the steps of SDM and have integrated the SDM process into daily practice. Although they are not alert to SDM perception and neglect the practice skills, they believe that perception of SDM is essential, and we can strengthen the perception of SDM, leading to positive attitudes toward and improving the practice and skill of SDM.(5)The SDM competency development for medical residents: During residency training, medical residents follow the instructions of the supervising (attending) physician. Although they learn to discuss patient problems with team members, their role in decision making is more passive [74]. A qualitative study showed that interns know different content and learn differently at rounds with or without an attending physician. Interns learned SDM with families from observing attending physicians’ communication and had not seen medical residents do it [76]. While residency training programs assess how medical residents acquire implementation of SDM [77], there is little research on how a resident’s contextual factors or the attending physician’s opinion can influence a resident’s opportunity to practice these skills. In exploring barriers to medical residents learning SDM implementation, studies have identified a lack of familiarity with SDM concepts, lack of feedback on communication skills, and lack of formal training education [78,79]. Our study observed that medical residents feel the importance and are aware of SD and RI aspects rather than the PP and ES aspects by IAA analysis. We should focus on improving the PP and ES aspects and keeping the SD and RI aspects for medical residents.(6)The SDM competency development for PGYs: The medical education system in Taiwan has been reformed in recent years: 6 years of medical school after senior high school graduation, including two years of clinical clerkship at hospitals. After passing the national examination for physicians, doctors have two years of post-graduate training, followed by the residency training in various specialties [80]. Several studies have documented insufficient training for SDM and patient-centered communication during graduate medical education [47,81]. The PGY training programs are non-specific and focus on the general medical training to improve the ability of patient-centered care. They belong to external stakeholders as compared to specialty-specific medical residents in the medical team. Their experiences of SDM implementation are sparse. Based on the NRM analysis of our study, the SDM concept and practice should be rooted and promoted in junior doctors, especially in implementing the aspect of perception.(7)Design talent cultivation programs for SDM competency development: The practice of SDM impacts patients, physicians, and hospitals alike. How can we better cultivate medical residents or PGYs to obtain ideal SDM implementation during their training and beyond? Several measures would improve physicians’ professionalism on SDM in preparing young doctors for this critical task; for example, clarifying SDM as a core clinical practice that needs planned teaching, confirming that attending physicians are adequately competent in fundamental elements of SDM, evaluating and providing feedback through direct observation, using OSCE or EPA exams to measure learning outcomes. It is critical that attending physicians themselves demonstrate contemporary understanding and competence for SDM through continuing medical education, grand rounds, on-the-job training, and other forms of continuing professional development.(8)Tailored sustainable training plans for different physician stakeholders: This study determined that the stakeholders should include senior physicians (attending physicians and medical residents) and junior physicians (PGYs), even though there are no common adoption paths based on all physicians as illustrated in Table 14. Therefore, there are three common adoption paths (PP→SD→RI; PP→ES→RI; PP→ES→SD→RI) based on the senior physicians (attending physicians and medical residents). The second suited adoption path is that the PP aspect improves the SD aspect and the SD aspect improves the RI aspect. The third suited adoption path is that the PP aspect influences the ES aspect and the ES aspect affects the RI aspect. The fourth suited adoption path is that the PP aspect affects the ES aspect, the ES aspect improves the SD aspect, and then the SD aspect improves the RI aspect, as shown in Table 14. The senior physicians hold the core role of SDM promotion in clinical practice and senior physicians have more opportunities to practice and use SDM. For both attending physicians and medical residents, more training for SDM is likely warranted. Based on our findings, the improvement of SDM perception and evaluation for senior physicians can enhance skills and practice of SDM, help clinicians engage in SDM, and initiate SDM conversations. Doctor–patient interactions can be more harmonious and appropriate decisions can be made together.(9)Comparing our results with those of other studies: In previous research on SDM training programs and surveys on physicians’ understanding of SDM, most only describe the training duration, training instruments, course assessment, teaching methods, content, trainees’ reflection, benefit or abilities obtained from SDM training, and facilitators or barriers for SDM promotion [3,4,47,49,73,74,78]. There is little evidence to suggest which SDM core competency training programs are most effective and which physicians’ SDM capabilities are priorities to require. This study uses the IAA-NRM process to investigate the critical aspects of SDM competency and the various aspects’ interactions and to create suitable development strategies.

## 5. Conclusions and Recommendations

### 5.1. Conclusions

SDM is defined as the interactive process by which doctors and patients make healthcare choices together [1]. The purpose of this study was to investigate the competency development of SDM tasks from diverse physicians’ opinions. There are five stages in the research design. The first stage summarizes the literature review and expert interviews to determine the core competency content. The research eventually identified four aspects, including perception assessment and practice (PP), execution process and skills (ES), physician–patient relationship and interaction (RI), and shared information and decision making (SD). The second stage was the survey process, which included the questionnaire design, data collection, and reliability tests. The third stage used the IAA approach to establish the importance and awareness of the aspects. The fourth stage developed the adoption path using the NRM approach. Finally, the last stage explored the adoption strategies for SDM competency development in all stakeholders’ opinions, including attending physicians, medical residents, and PGYs.

There were several study findings as follows: There is no common suitable path for the viewpoints of the attending physicians, medical residents, and PGYs. However, there are three common suitable paths in the opinions of the senior physicians (attending physicians and medical residents). The first path starts from the PP to SD aspect and then the RI aspect. The second path is from the PP to ES and then the RI aspect. The third path is from PP to ES, ES to SD, and then the RI aspect (PP→SD→RI; PP→ES→RI; PP→ES→SD→RI). All these critical paths start with the PP aspect. For senior physicians, a core requirement of the SDM implementation is to improve knowledge and awareness.

### 5.2. Academic Contributions

We used the IAA method to examine the status of importance and awareness for three physicians (attending physicians, medical residents, and PGYs) and adopted the NRM approach to estimate the network relation structure by the DEMATEL methodology. The combined IAA-NRM method supplies a helpful tool for hospital administrators and directors of medical education to understand the various physicians’ opinions on clinicians’ SDM competency strategies and to identify suitable paths for SDM implementation. This study utilized survey methodology to examine the needs of various physicians in Taiwan. Physicians’ preferences in other countries may not be the same; therefore, the proper SDM promotion strategies assumed by other countries will not be the same either. However, the perspectives areas of concern and different improvement strategies of diverse physicians in this study can provide meaningful information and valuable suggestions.

### 5.3. Study Limitations

This study has some limitations. First, the research results based on one medical center and one regional hospital may not be generalizable to other medical institutions. Second, the study analyzed various physicians (attending physicians, medical residents, and PGYs). The viewpoints of PGYs might impact the research results of the IAA-NRM approach. Multiple factors can affect a physician’s experience with the performance of SDM, such as gender, seniority, persons per se, workplaces, social services, and various specialties [72,76,82,83]. However, a number of varying confounders were not controlled as a matched study population to evaluate the influence of SDM implementation on the outcome and this may have led to some bias in need for competency development. Third, the participants of this study were selected through snowball sampling. Thus, our results should be interpreted cautiously, as the sample does not necessarily represent all Taiwanese physicians’ viewpoints.

### 5.4. Future Studies

While each institution has unique characteristics, organizational cultures, constraints, and different physician stakeholders may play critical roles in facilitating SDM implementation. We should respect the interests and demands of the main stakeholders and achieve their best interests through communication. The authors selected three physician stakeholders (attending physicians, residents, and PGYs) from the relevant literature as the leading subjects for this study. Future research could focus on the perspectives of other multi-level stakeholders, such as other healthcare providers, patients, and their families. In addition, the current survey for physicians’ SDM competencies in general conditions has not focused on particular diseases or certain clinical scenarios (e.g., loss of autonomous ability to rely on family members or primary caregivers to make decisions), but the future study could be conducted to compare physicians’ SDM competencies for different specialists or disease-specific situations.

## Figures and Tables

**Figure 1 healthcare-10-01844-f001:**
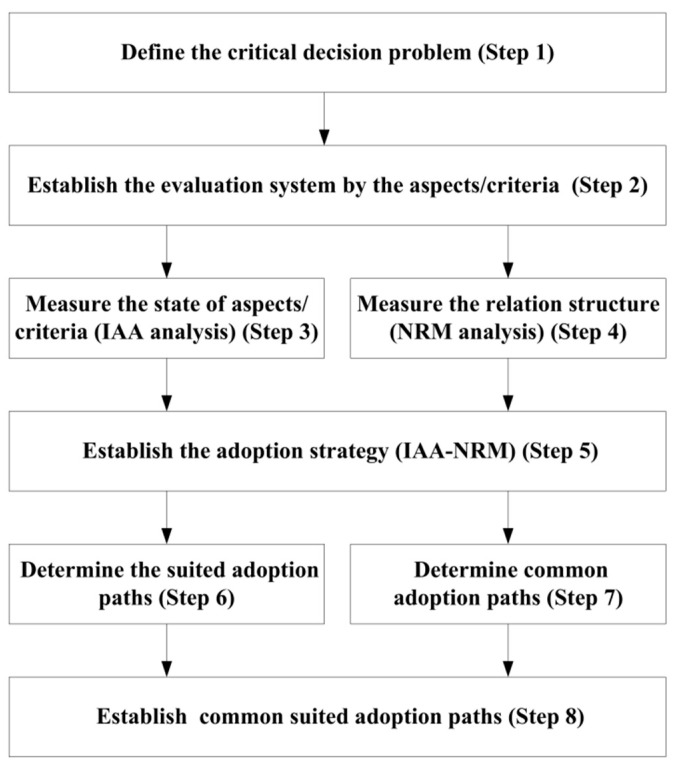
The SDM competency development based on the IAA-NRM approach.

**Figure 2 healthcare-10-01844-f002:**
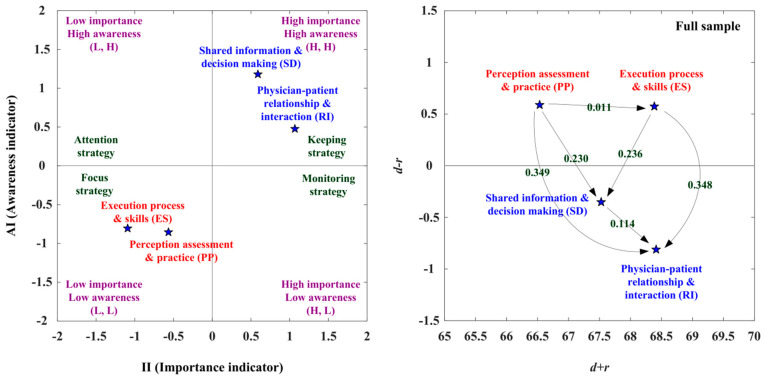
The IAA-NRM map of SDM competency development.

**Figure 3 healthcare-10-01844-f003:**
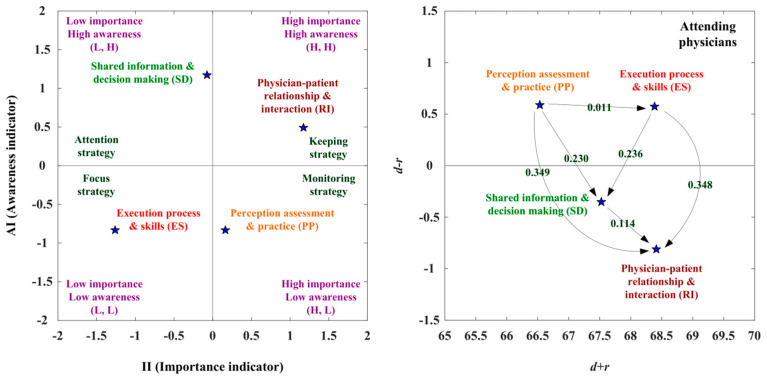
The SDM competency development map for attending physicians.

**Figure 4 healthcare-10-01844-f004:**
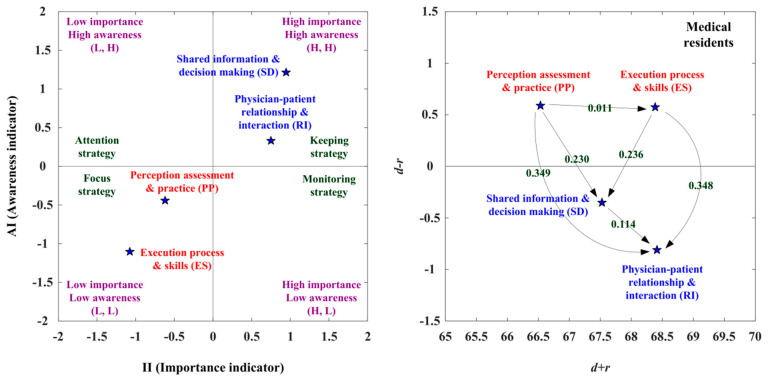
The SDM competency development map for medical residents.

**Figure 5 healthcare-10-01844-f005:**
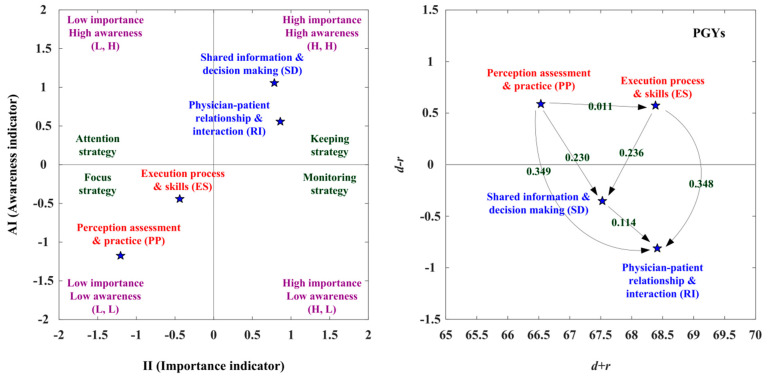
The SDM competency development map for PGYs.

**Table 2 healthcare-10-01844-t002:** The analysis of reliability (Cronbach α).

Aspects/Criteria	Alpha	Result
Importance indicator	0.969	High
Awareness indicator	0.964	High
Aspects of evaluation system	0.947	High

Note: Cronbach alpha α-value: α ≤ 0.35 is low reliability; 0.35 < α < 0.7 is middle reliability; α ≥ 0.7 is high reliability.

**Table 3 healthcare-10-01844-t003:** The adoption strategy of SDM competency development.

Aspects	IAA			NRM			AS
	II	AI	(II, AI)	*d* + *r*	*d* − *r*	(R, D)	
Perception assessment and practice (PP)	−0.565	−0.854	L, L	66.534	0.589	D (+,+)	C
Execution process and skills (ES)	−1.092	−0.805	L, L	68.384	0.574	D (+,+)	C
Physician–patient relationship and interaction (RI)	1.067	0.478	H, H	68.413	−0.811	ID (+,−)	A
Shared information and decision making (SD)	0.590	1.181	H, H	67.526	−0.352	ID (+,−)	A

Notes: The adoption strategies (AS) include four types: adoption strategy A (keeping strategy), adoption strategy B (attention strategy), adoption strategy C (focus strategy), and adoption strategy D (monitoring strategy). “+” means direct development and “−” means indirect development.

**Table 4 healthcare-10-01844-t004:** The suited adoption paths of SDM competency development.

	II (Importance Indicator)	AI (Awareness Indicator)
Rank	RI(1) > SD(2) > PP(3) > ES(4)	SD(1) > RI(2) > ES(3) > PP(4)
Available paths	1. PP(3)→RI(1) {N} 2. PP(3)→SD(2)→RI(1) {N} 3. PP(3)→ES(4)→RI(1) {Y} 4. PP(3)→ES(4)→SD(2)→RI(1) {Y}	1. PP(4)→RI(2) {N} 2. PP(4)→SD(1)→RI(2) {Y} 3. PP(4)→ES(3)→RI(2) {N} 4. PP(4)→ES(3)→SD(1)→RI(2) {Y}
Suited adoption paths	4. PP→ES→SD→RI	

**Table 5 healthcare-10-01844-t005:** The common adoption paths of competency development for attending physicians and medical residents.

	II (Importance Indicator)	AI (Awareness Indicator)	Suited Adoption Paths
Attending physicians			
Rank	RI(1) > PP(2) > SD(3) > ES(4)	SD(1) > RI(2) > PP(3) = ES(3)	
Available paths	1. PP(2)→RI(1) {N} 2. PP(2)→SD(3)→RI(1) {Y} 3. PP(2)→ES(4)→RI(1) {Y} 4. PP(2)→ES(4)→SD(3)→RI(1) {Y}	1. PP(3)→RI(2) {N} 2. PP(3)→SD(1)→RI(2) {Y} 3. PP(3)→ES(3)→RI(2) {Y} 4. PP(3)→ES(3)→SD(1)→RI(2) {Y}	2. PP→SD→RI 3. PP→ES→RI 4. PP→ES→SD→RI
Medical residents			
Rank	SD(1) > RI(2) > PP(3) > ES(4)	SD(1) > RI(2) > PP(3) > ES(4)	
Available paths	1. PP(3)→RI(1) {N} 2. PP(3)→SD(1)→RI(2) {Y} 3. PP(3)→ES(4)→RI(2) {Y} 4. PP(3)→ES(4)→SD(1)→RI(2) {Y}	1. PP(3)→RI(2) {N} 2. PP(3)→SD(1)→RI(2) {Y} 3. PP(3)→ES(4)→RI(2) {Y} 4. PP(3)→ES(4)→SD(1)→RI(2) {Y}	2. PP→SD→RI 3. PP→ES→RI 4. PP→ES→SD→RI
Common adoption paths	2. PP→SD→RI 3. PP→ES→RI 4. PP→ES→SD→RI		Common suited adoption paths
			2. PP→SD→RI 3. PP→ES→RI 4. PP→ES→SD→RI

**Table 6 healthcare-10-01844-t006:** The adoption strategy of SDM competency development for attending physicians.

Aspects	IAA			NRM			AS
	II	AI	(II, AI)	*d* + *r*	*d* − *r*	(R, D)	
Perception assessment and practice (PP)	0.163	−0.832	H, L	66.534	0.589	D (+,+)	D
Execution process and skills (ES)	−1.262	−0.832	L, L	68.384	0.574	D (+,+)	C
Physician–patient relationship and interaction (RI)	1.173	0.492	H, H	68.413	−0.811	ID (+,−)	A
Shared information and decision making (SD)	−0.074	1.172	L, H	67.526	−0.352	ID (+,−)	B

Notes: The adoption strategies (AS) include four types: adoption strategy A (keeping strategy), adoption strategy B (attention strategy), adoption strategy C (focus strategy), and adoption strategy D (monitoring strategy). “+” means direct development and “−” means indirect development.

**Table 7 healthcare-10-01844-t007:** The suited adoption paths of SDM competency development for attending physicians.

	II (Importance Indicator)	AI (Awareness Indicator)
Rank	RI(1) > PP(2) > SD(3) > ES(4)	SD(1) > RI(2) > PP(3) = ES(3)
Available paths	1. PP(2)→RI(1) {N} 2. PP(2)→SD(3)→RI(1) {Y} 3. PP(2)→ES(4)→RI(1) {Y} 4. PP(2)→ES(4)→SD(3)→RI(1) {Y}	1. PP(3)→RI(2) {N} 2. PP(3)→SD(1)→RI(2) {Y} 3. PP(3)= ES(3)→RI(2) {Y} 4. PP(3)= ES(3)→SD(1)→RI(2) {Y}
Suited adoption paths	2. PP→SD→RI 3. PP→ES→RI 4. PP→ES→SD→RI	

**Table 8 healthcare-10-01844-t008:** The adoption strategy of SDM competency development for medical residents.

Aspects	IAA			NRM			AS
	II	AI	(II, AI)	*d* + *r*	*d* − *r*	(R, D)	
Perception assessment and practice (PP)	−0.620	−0.442	L, L	66.534	0.589	D (+,+)	C
Execution process and skills (ES)	−1.076	−1.104	L, L	68.384	0.574	D (+,+)	C
Physician–patient relationship and interaction (RI)	0.750	0.331	H, H	68.413	−0.811	ID (+,−)	A
Shared information and decision making (SD)	0.946	1.215	H, H	67.526	−0.352	ID (+,−)	A

Notes: The adoption strategies (AS) include four types: adoption strategy A (keeping strategy), adoption strategy B (attention strategy), adoption strategy C (focus strategy), and adoption strategy D (monitoring strategy). “+” means direct development and “−” means indirect development.

**Table 9 healthcare-10-01844-t009:** The suited adoption paths of SDM competency development for medical residents.

	II (Importance Indicator)	AI (Awareness Indicator)
Rank	SD(1) > RI(2) > PP(3) > ES(4)	SD(1) > RI(2) > PP(3) > ES(4)
Available paths	1. PP(3)→RI(2) {N} 2. PP(3)→SD(1)→RI(2) {Y} 3. PP(3)→ES(4)→RI(2) {Y} 4. PP(3)→ES(4)→SD(1)→RI(2) {Y}	1. PP(3)→RI(2) {N} 2. PP(3)→SD(1)→RI(2) {Y} 3. PP(3)→ES(4)→RI(2) {Y} 4. PP(3)→ES(4)→SD(1)→RI(2) {Y}
Suited adoption paths	2. PP→SD→RI 3. PP→ES→RI 4. PP→ES→SD→RI	

**Table 10 healthcare-10-01844-t010:** The adoption strategy of SDM competency development for PGYs.

Aspects	IAA			NRM			AS
	II	AI	(II, AI)	*d* + *r*	*d* − *r*	(R, D)	
Perception assessment and practice (PP)	−1.205	−1.174	L, L	66.534	0.589	D (+,+)	C
Execution process and skills (ES)	−0.440	−0.441	L, L	68.384	0.574	D (+,+)	C
Physician–patient relationship and interaction (RI)	0.861	0.558	H, H	68.413	−0.811	ID (+,−)	A
Shared information and decision making (SD)	0.784	1.057	H, H	67.526	−0.352	ID (+,−)	A

Notes: The adoption strategies (AS) include four types: adoption strategy A (keeping strategy), adoption strategy B (attention strategy), adoption strategy C (focus strategy), and adoption strategy D (monitoring strategy). “+” means direct development and “−” means indirect development.

**Table 11 healthcare-10-01844-t011:** The suited adoption paths of competency development for PGYs.

	II (Importance Indicator)	AI (Awareness Indicator)
Rank	RI(1) > SD(2) > ES(3) > PP(4)	SD(1) > RI(2) > ES(3) > PP(4)
Available paths	1. PP(4)→RI(1) {N} 2. PP(4)→SD(2)→RI(1) {N} 3. PP(4)→ES(3)→RI(1) {N} 4. PP(4)→ES(3)→SD(2)→RI(1) {N}	1. PP(4)→RI(2) {N} 2. PP(4)→SD(1)→RI(2) {Y} 3. PP(4)→ES(3)→RI(2) {N} 4. PP(4)→ES(3)→SD(1)→RI(2) {Y}
Suited adoption paths	-	

**Table 12 healthcare-10-01844-t012:** Common adoption paths for II (importance indicator).

	II (Importance Indicator)
Attending physicians	
Rank	RI(1) > PP(2) > SD(3) > ES(4)
Available paths	1. PP(2)→RI(1) {N} 2. PP(2)→SD(3)→RI(1) {Y} 3. PP(2)→ES(4)→RI(1) {Y} 4. PP(2)→ES(4)→SD(3)→RI(1) {Y}
Medical residents	
Rank	SD(1) > RI(2) > PP(3) > ES(4)
Available paths	1. PP(3)→RI(2) {N} 2. PP(3)→SD(1)→RI(2) {Y} 3. PP(3)→ES(4)→RI(2) {Y} 4. PP(3)→ES(4)→SD(1)→RI(2) {Y}
PGYs	
Rank	RI(1) > SD(2) > ES(3) > PP(4)
Available paths	1. PP(4)→RI(1) {N} 2. PP(4)→SD(2)→RI(1) {N} 3. PP(4)→ES(3)→RI(1) {N} 4. PP(4)→ES(3)→SD(2)→RI(1) {N}
Common adoption paths	-

**Table 13 healthcare-10-01844-t013:** Common adoption paths for AI (awareness indicator).

	AI (Awareness Indicator)
Attending physicians	
Rank	SD(1) > RI(2) > PP(3) =ES(3)
Available paths	1. PP(3)→RI(2) {N} 2. PP(3)→SD(1)→RI(2) {Y} 3. PP(3)→ES(3)→RI(2) {Y} 4. PP(3)→ES(3)→SD(1)→RI(2) {Y}
Medical residents	
Rank	SD(1) > RI(2) > PP(3) > ES(4)
Available paths	1. PP(3)→RI(2) {N} 2. PP(3)→SD(1)→RI(2) {Y} 3. PP(3)→ES(4)→RI(2) {Y} 4. PP(3)→ES(4)→SD(1)→RI(2) {Y}
PGYs	
Rank	SD(1) > RI(2) > ES(3) > PP(4)
Available paths	1. PP(4)→RI(2) {N} 2. PP(4)→SD(1)→RI(2) {Y} 3. PP(4)→ES(3)→RI(2) {N} 4. PP(4)→ES(3)→SD(1)→RI(2) {Y}
Common adoption paths	2. PP(4)→SD(1)→RI(2) 4. PP(4)→ES(3)→SD(1)→RI(2)

**Table 14 healthcare-10-01844-t014:** The common suited adoption paths for SDM competency development of SDM tasks (attending physicians, medical residents, and PGYs).

	II (Importance Indicator)	AI (Awareness Indicator)	Suited Adoption Paths
**Senior physician** -Attending physicians			
Rank	RI(1) > PP(2) > SD(3) > ES(4)	SD(1) > RI(2) > PP(3) = ES(3)	
Available paths	1. PP(2)→RI(1){N} 2. PP(2)→SD(3)→RI(1) {Y} 3. PP(2)→ES(4)→RI(1) {Y} 4. PP(2)→ES(4)→SD(3)→RI(1) {Y}	1. PP(3)→RI(2){N} 2. PP(3)→SD(1)→RI(2) {Y} 3. PP(3)= ES(3)→RI(2) {Y} 4. PP(3)= ES(3)→SD(1)→RI(2) {Y}	2. PP→SD→RI 3. PP→ES→RI 4. PP→ES→SD→RI
**Senior Physician** -Medical residents			
Rank	SD(1) > RI(2) > PP(3) > ES(4)	SD(1) > RI(2) > PP(3) > ES(4)	
Available paths	1. PP(3)→RI(2) {N} 2. PP(3)→SD(1)→RI(2) {Y} 3. PP(3)→ES(4)→RI(2) {Y} 4. PP(3)→ES(4)→SD(1)→RI(2) [Y]	1. PP(3)→RI(2) {N} 2. PP(3)→SD(1)→RI(2) {Y} 3. PP(3)→ES(4)→RI(2) {Y} 4. PP(3)→ES(4)→SD(1)→RI(2) {Y}	2. PP→SD→RI 3. PP→ES→RI 4. PP→ES→SD→RI
Common adoption paths (attending physicians and medical residents)	2. PP→SD→RI 3. PP→ES→RI 4. PP→ES→SD→RI	2. PP→SD→RI 3. PP→ES→RI 4. PP→ES→SD→RI	Common suited adoption paths
			2. PP→SD→RI 3. PP→ES→RI 4. PP→ES→SD→RI
**Junior physician** -PGYs			
Rank	RI(1) > SD(2) > ES(3) > PP(4)	SD(1) > RI(2) > ES(3) > PP(4)	
Available paths	1. PP(4)→RI(1) {N} 2. PP(4)→SD(2)→RI(1) {N} 3. PP(4)→ES(3)→RI(1) {N} 4. PP(4)→ES(3)→SD(2)→RI(1) {N}	1. PP(4)→RI(2) {N} 2. PP(4)→SD(1)→RI(2) {Y} 3. PP(4)→ES(3)→RI(2) {N} 4. PP(4)→ES(3)→SD(1)→RI(2) {Y}	-
Common adoption paths (attending physicians, medical residents, and PGYs)	-	2. PP→SD→RI 4. PP→ES→SD→RI	Common suited adoption paths
			-

## Data Availability

The data presented in this study are available on reasonable request from the corresponding author.

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
