# Peer review of "Establishing the Competency Development and Talent Cultivation Strategies for Physician-Patient Shared Decision-Making Competency Based on the IAA-NRM Approach"

_healthcare, 2022, doi:10.3390/healthcare10101844_

Round 1

Reviewer 1 Report

This study presents an interesting research model that integrates Importance-performance analysis with the DEMATEL approach and applies it to the topic of shared decision-making. Addition, the case subjects include physicians, medical residents, and doctors in post-graduate for 27 years. Therefore, the authors need to pay attention to how to express conciseness, so that more general readers can easily understand the content and values of the study. Some questions or suggestions are as follows:

1. Compared to “Important-Awareness Analysis”, the “Importance-Performance Analysis (IPA)" is the most common method, why not use the IPA method directly? In other words, the authors needed to wonder whether the study clearly expressed "Importance-Awareness", especially "Awareness".

2. There is too much content in the introduction, and it is suggested that the author should focus on the background, problems, research status, and gaps, research design and process. Among them, the problems and research status and gaps are recommended to strengthen!

3.    When using an abbreviation for the first time, the author should first write the full name (abbreviation), that is, define the abbreviation word first. For example, SHARE.

4. Regarding Materials and Methods, the structure and content of this section are very important, and the reader should be able to easily understand the study design and process, and the corresponding method calculations. However, the current version shows complex content, especially sections 2.2 to 2.6. This question is easy makes it difficult for the reader to understand. Individuals suggest that the results should be presented uniformly in the results section. In other words, this section should clearly express the construction of indicator models, the introduction and steps of methods, and the design and analysis of studies. The results regarding the application of this method should be presented in the next section. It is recommended that readers find several articles on the application of multi-criteria decision-making methods in shared decision-making. In addition, there are too many tables and diagrams, and it is not easy for readers to read.

5. Both the aspects and criteria in Table 1 should mark the source of the cited literature.

6. Why there are two thesis review numbers in this study suggests that the authors need to briefly state.

7. In Table 3, What is II, AI, MI, MI, SI, MA, SA, it is recommended to have the full name before the abbreviation. For example, the mean importance level (MI).

8. The mathematical equations of the DEMATEL method are expressed incorrectly, for example, in equation (2), D=[xD= [xD= [xij]. It is recommended to read Tzeng G.H.-related articles. The scholar's research on the application of the DEMATEL method is very rich and helpful, which can help the authors to present this section more concisely and easily understood.

9. In equation (3), the matrix I should be clear.

10. The format of d, r, d+r, and d-r should be uniform. For example, {d+r} (Table 9), d+r (Table 11).

11. In fig. 3, PP with ES affecting the RI arrows should be drawn, at least ES should be drawn.

12. The symbol expressed in the ranking should be >. One of the cases is on page 364. The author read several more SCI/SSCI articles.

13. [  ] is the citation format for this journal. It is recommended to express it in a different way because it is easy to confuse the reader. One of the cases is on page 364.

14. The decision analysis model proposed in this study is applied in three different roles, namely A, B, and C. In the discussion section, the author needs to think about how the sub can express it more concisely and easily understood.

The above questions may be listed by the author, but it is still recommended that the author re-examine the structure and content of this article from the perspective of the general reader. In the end, this study is a very interesting and corresponding decision analysis model. 

Author Response

Response to Reviewer 1 Comments

Point 1: Compared to “Important-Awareness Analysis”, the “Importance-Performance Analysis (IPA)" is the most common method, why not use the IPA method directly? In other words, the authors needed to wonder whether the study clearly expressed "Importance-Awareness", especially "Awareness".

Response 1: Thank you for your comments. The IIA approach was based on the importance-performance analysis presented by Martilla and James [Ref 51], but “awareness” replaced “performance” in this study. Several studies demonstrated an evaluation model using similar IAA methods to explore the importance and relationship between factors [Ref 52-55]. The reason for choosing "awareness" in this study is that literacy is a crucial measure of a population’s education. Literacy is the key to improving peoples' capabilities and accomplishing many other rights [Ref 26]. The definition of literacy in UNESCO’s 2030 Agenda for Sustainability Development Goals is “a means of identification, understanding, interpretation, creation, and communication in an increasingly digital, text-mediated, information-rich and fast-changing world.” [https://en.unesco.org/creativity/sites/creativity/files/247785en.pdf] In most clinical conditions, the clinician is always the initiator of medical decisions and should be familiar with participating in the SDM process. Physicians' higher awareness of the uncertainty about treatment options could improve their communication performance in decision-making [Ref 27]. However, most healthcare professionals enter the clinical setting without adequate SDM skills and may be unfamiliar with the perception of SDM. This study defines SDM; outlines the concepts, affordances, and common applications, and conceptualizes “SDM” as a literacy practice for medical education and impacting patient-centered care settings. Thus, this study aimed to examine physicians' awareness of SDM literacy and to identify which SDM competencies are important to develop. So, we use the importance-awareness analysis as our assessment method. We have modified the Introduction and Materials & Methods section (see line 88-96, line 244-249).

Point 2: There is too much content in the introduction, and it is suggested that the author should focus on the background, problems, research status, and gaps, research design and process. Among them, the problems and research status and gaps are recommended to strengthen!.

Response 2: Thank you for your comments. The paper has been modified by removing a lot of redundant descriptions in the Introduction section. We have enhanced the problems, research status, and gaps. We have modified the Introduction section.

Point 3: When using an abbreviation for the first time, the author should first write the full name (abbreviation), that is, define the abbreviation word first. For example, SHARE.

Response 3: Thank you for your suggestions. We have added the full name of SHARE (see line 86). We also have a list of abbreviations at the end of the manuscript (see line 857-867).

Point 4: Regarding Materials and Methods, the structure and content of this section are very important, and the reader should be able to easily understand the study design and process, and the corresponding method calculations. However, the current version shows complex content, especially sections 2.2 to 2.6. This question is easy makes it difficult for the reader to understand. Individuals suggest that the results should be presented uniformly in the results section. In other words, this section should clearly express the construction of indicator models, the introduction and steps of methods, and the design and analysis of studies. The results regarding the application of this method should be presented in the next section. It is recommended that readers find several articles on the application of multi-criteria decision-making methods in shared decision-making. In addition, there are too many tables and diagrams, and it is not easy for readers to read.

Response 4: We apologize for the confusion we caused. To avoid ambiguity, we have rewritten our statements in the Materials and Methods section. The results have been presented uniformly in the Results section. The number of final tables has been reduced from 22 to 14. The number of final figures had declined from 7 to 5. Other forms are displayed in the Supplementary File.

Point 5: Both the aspects and criteria in Table 1 should mark the source of the cited literature.

Response 5: Thank you for your comment. We have added the source of the cited references into Table 1, as the reviewer suggested (see page 6).

Point 6: Why there are two thesis review numbers in this study suggests that the authors need to briefly state.

Response 6: Thank you for your comment. IRB number: 202200716B0 approved the initial enrollment of 100 participants, but later on, it was necessary to increase the sample size to improve the reliability, so a revised enrollment of 150 participants was approved by IRB number 202200716B0C501. Please see the attached file about IRB-approved documents.

Point 7: In Table 3, What is II, AI, MI, MI, SI, MA, SA, it is recommended to have the full name before the abbreviation. For example, the mean importance level (MI).

Response 7: Thank you for pointing this out. Table 3 has been changed to Table S1 in the revised manuscript. We have modified Table S1 as the reviewer suggested (see Supplement file, page 1, Table S1, Note 2, line 13-14).

Point 8: The mathematical equations of the DEMATEL method are expressed incorrectly, for example, in equation (2), D=[xD= [xD= [xij]. It is recommended to read Tzeng G.H.-related articles. The scholar's research on the application of the DEMATEL method is very rich and helpful, which can help the authors to present this section more concisely and easily understood.

Response 8: Thank you for pointing this out. The manuscript already modifies Equation 3 based on the Tzeng G.H.-related article. [Ref 60: Lin, C. L., Shih, Y. H., Tzeng, G. H., and Yu, H. C. (2016). A service selection model of digital music platform by using a novel MCDM technique. Applied Soft Computing, 48, 385–403.]. The algorithm for Equation (2) in their study and our research is shown in the attached file (see line 296-299 in the revised manuscript).

Point 9: In equation (3), the matrix I should be clear..

Response 9: Thank you for pointing this out. The manuscript already modifies Equation 3 based on the Tzeng G.H.-related article [Ref 60]. The algorithm for Equation (3) in their study and our research is shown in the attached file (see line 303 in the revised manuscript).

Point 10: The format of d, r, d+r, and d-r should be uniform. For example, {d+r} (Table 9), d+r (Table 11)..

Response 10: The reviewer is correct. Table 9 has been changed to Table S7 in the revised manuscript. We have modified Table 9 as the reviewer suggested (see Supplement file, page 3, Table S7).

Point 11: In fig. 3, PP with ES affecting the RI arrows should be drawn, at least ES should be drawn.

Response 11: Thank you for your comment. Figure 3 has been changed to Figure S2 in the revised manuscript. This information has now been presented explicitly in Figure S2 (see Supplement file, page 4, Figure S2).

Point 12: The symbol expressed in the ranking should be >. One of the cases is on page 364. The author read several more SCI/SSCI articles.

Response 12: The reviewer is correct. We have made revisions accordingly.

Point 13: [  ] is the citation format for this journal. It is recommended to express it in a different way because it is easy to confuse the reader. One of the cases is on page 364.

Response 13: We thank the reviewer for this excellent suggestion. We change the [ ] symbol to ( ) per the reviewer’s suggestions.

Point 14: The decision analysis model proposed in this study is applied in three different roles, namely A, B, and C. In the discussion section, the author needs to think about how the sub can express it more concisely and easily understood.

Response 14: Thank you for your comment. We have added the statements of the subsections in the Discussion section (marked in red color). We have modified the Discussion section as the reviewer suggested (line 657, 668, 684-686, 709, 725, 741, 754, 766).

Once again, thank you very much for your comments and suggestions.

Thanks to the reviewers for the thoughtful and thorough review.

Hopefully, we have addressed all of your concerns.

Reviewer 2 Report

This paper is developed within the healthcare shared decision-making framework, in which patients and medical staff work together to make decisions about patients treatments. The aim of this study is to investigate the competency development of shared decision-making tasks taking into account the opinions of physicians.

After a detailed review, I consider that this is an interesting an well-written paper, which might be of interest for the readers of the Healthcare journal. Before accepting it, I consider that the authors should revise some issues which are pointed in the attached pdf file.

Author Response

Response to Reviewer 2 Comments

Point 1: I consider that Figure 1 should be better introduced and explained..

Response 1: We thank the reviewer for this excellent suggestion. We have added the explanations for Figure 1 as the reviewer’s suggestion (see line 121-122, line 125-142).

Point 2: In lines 123-124 the authors mention Excel and MATLAB. I consider that they should explain which versions were used, and if any toolbox was contemplated.

Response 2: Thank you for pointing this out. We added the information about versions of Excel and MATLAB (see line 144-145).

Point 3: I consider that the authors should add a subsection reviewing some recent applications using shared decision-making in the healthcare area.

Response 3: We thank the reviewer for this excellent suggestion. As the reviewer suggested, we have added a subsection in the Introduction section on the applications of shared decision-making in healthcare settings (see line 57-62).

Point 4: I recommend the authors to explain in more detail figures 3 and 4.

Response 4: Thank you for your comment. In the revised manuscript, Figure 3 has been changed to Figure S2, and Figure 4 has been changed to Figure 2. As the reviewer suggested, we have added more detailed explanations for Figures 2 and S2 (see line 338-351; Supplement File: page 3, line 64-66).

Point 5: The discussion section should clearly determine what is the main contribution of the article compared to other studies or similar works in the related field of study. Authors need to pay special attention to this comparison and highlight the relevance of their contributions.

Response 5: We thank the reviewer for this excellent suggestion. We have modified the discussion section (see line 784-792).

Point 6: I recommend the authors to add a paragraph at the end of the introduction section explaining how the paper is organised.

Response 6: Thank you for pointing this out. We have added a paragraph at the end of the introduction section to explain how the paper is organized (see line 109-119).

Point 7: I understand that the use of acronyms is helpful to reduce de length of the paper. However, it can make it very difficult to follow it. I consider that it would be helpful if you could have a list of abbreviations at the end or at the beginning of the manuscript.

Response 7: Thank you for your nice reminder. We have a list of abbreviations at the end of the manuscript (see line 857-867).

Once again, thank you very much for your comments and suggestions.

Thanks to the reviewers for the thoughtful and thorough review.

Hopefully, we have addressed all of your concerns.

Round 2

Reviewer 1 Report

Thanks to the authors for their careful revision of this article.